# Influence of Lymphatic, Microvascular and Perineural Invasion on Oncological Outcome in Patients with Neuroendocrine Tumors of the Small Intestine

**DOI:** 10.3390/cancers16020305

**Published:** 2024-01-11

**Authors:** Frederike Butz, Agata Dukaczewska, Catarina Alisa Kunze, Janina Maren Krömer, Lisa Reinhard, Henning Jann, Uli Fehrenbach, Charlotte Friederieke Müller-Debus, Tatiana Skachko, Johann Pratschke, Peter E. Goretzki, Martina T. Mogl, Eva Maria Dobrindt

**Affiliations:** 1Department of Surgery, Campus Charité Mitte|Campus Virchow-Klinikum, Charité–Universitätsmedizin Berlin, Corporate Member of Freie Universität Berlin and Humboldt-Universität zu Berlin, 10117 Berlin, Germany; 2Department of Pathology, Charité–Universitätsmedizin Berlin, Corporate Member of Freie Universität Berlin and Humboldt-Universität zu Berlin, 10117 Berlin, Germany; 3Department of Hepatology and Gastroenterology, Campus Charité Mitte|Campus Virchow-Klinikum, Charité–Universitätsmedizin Berlin, Corporate Member of Freie Universität Berlin and Humboldt-Universität zu Berlin, 10117 Berlin, Germany; 4Department of Radiology, Charité–Universitätsmedizin Berlin, Corporate Member of Freie Universität Berlin and Humboldt-Universität zu Berlin, 10117 Berlin, Germany

**Keywords:** small-intestinal neuroendocrine tumors, microvascular invasion, lymphatic invasion, perineural invasion, oncological outcome

## Abstract

**Simple Summary:**

Lymphatic (LI), microvascular (VI) and perineural invasion (PnI) have been determined as indicators for aggressive tumor behavior and worse outcome in many solid tumors. In neuroendocrine tumors of the small intestine (siNET), some prognosis-defining factors have been well established, but the role of LI, VI and PnI remains incompletely understood so far. The aim of our retrospective study was to elucidate the role of lymphatic, microvascular and perineural invasion in the oncological outcome in siNET. We found that lymphatic, microvascular and perineural invasion led to earlier disease recurrence and postoperative disease progression. We therefore promote the routine description of these histopathological parameters for considerations on adjuvant treatment and follow-up.

**Abstract:**

For the histopathological work-up of resected neuroendocrine tumors of the small intestine (siNET), the determination of lymphatic (LI), microvascular (VI) and perineural (PnI) invasion is recommended. Their association with poorer prognosis has already been demonstrated in many tumor entities. However, the influence of LI, VI and PnI in siNET has not been sufficiently described yet. A retrospective analysis of all patients treated for siNET at the ENETS Center of Excellence Charité–Universitätsmedizin Berlin, from 2010 to 2020 was performed (*n* = 510). Patients who did not undergo primary resection or had G3 tumors were excluded. In the entire cohort (*n* = 161), patients with LI, VI and PnI status had more distant metastases (48.0% vs. 71.4%, *p* = 0.005; 47.1% vs. 84.4%, *p* < 0.001; 34.2% vs. 84.7%, *p* < 0.001) and had lower rates of curative surgery (58.0% vs. 21.0%, *p* < 0.001; 48.3% vs. 16.7%, *p* < 0.001; 68.4% vs. 14.3%, *p* < 0.001). Progression-free survival was significantly reduced in patients with LI, VI or PnI compared to patients without. This was also demonstrated in patients who underwent curative surgery. Lymphatic, vascular and perineural invasion were associated with disease progression and recurrence in patients with siNET, and these should therefore be included in postoperative treatment considerations.

## 1. Introduction

Although small-intestinal neuroendocrine tumors (siNET) belong to a heterogeneous group of rare neoplasms, their incidence has been reported to be rising [1]. Despite the fact that most tumors are diagnosed in locally advanced or metastatic stages, patients suffering from this disease still face a relatively good prognosis. While complete tumor resection remains the only curative therapy, it is largely established that cytoreductive therapy in the sense of debulking surgery is beneficial for patients where complete resection cannot be achieved [2,3]. For the histopathological work-up of resected specimens, the determination of lymphatic (LI), vascular (VI) and perineural sheath (PnI) invasion is recommended in addition to the indication of proliferation-based grading, histopathological classification and tumor and lymph node stage [4,5]. They describe the presence of cancer cells in lymphatic vessels (LI), blood vessels (VI) or the perineural sheath (PnI). The association of L, V and Pn invasion with poorer prognosis has already been demonstrated in many tumor entities [6,7,8,9]. Additionally, there has been evidence for the involvement of LI, VI and PnI in the development of metastatic disease and disease recurrence in neuroendocrine neoplasms; VI, for example, has been determined as an independent risk factor for disease recurrence in pancreatic neuroendocrine tumors [10]. Additionally, LI has been demonstrated to be associated with lymph node metastases in appendiceal and rectal neuroendocrine tumors [11,12]. While some outcome- and prognosis-defining factors have already been established in siNET, the influence of lymphatic, microvascular and perineural invasion has not been sufficiently described yet.

Therefore, we aimed at investigating the association of LI, VI and PnI with the oncological outcome in siNET patients undergoing surgery.

## 2. Materials and Methods

The Charité Comprehensive Cancer Center database was searched for patients treated for siNET at the European Neuroendocrine Tumor Society (ENETS) Center of Excellence at the Charité–Universitätsmedizin Berlin, Germany, between January 2010 and December 2020. A total of *n* = 510 patients was identified. The following exclusion criteria were defined: non-performed surgical primary resection (*n* = 327, of which *n* = 93 had received other surgical interventions, *n* = 13 had endoscopic removal of the primary, *n* = 166 had advanced metastatic disease and *n* = 57 had other reasons for non-performed primary resection including, amongst others, patients’ will and the extent of comorbidities), age under 18 years (*n* = 2) and G3 tumors (*n* = 11). Moreover, cases in which neither L, V nor Pn invasion could be recalled from the histopathological record were excluded (*n* = 8). Medical records were reviewed retrospectively to collect epidemiological and clinicopathological data including age, sex, tumor stage based on the 8th edition of the UICC classification of malignant tumors and tumor grading as well as TNM classification according to the World Health Organization (WHO) grading system [13]. Histopathological reports were reviewed for L, V and Pn invasion. In addition, details about resection margins were collected; therefore, possible residual tumor mass of non-resected tumor tissue was not considered for R status. Surgery was considered curative if all macroscopic tumor mass (including lymph node or distant metastases) was removed. Patients were followed-up as recommended by the ENETS consensus guidelines [14]. Recurrence or progression of the disease were determined based on clinical and radiological assessment in accordance with the response evaluation criteria in solid tumors (RECIST). Approval of the study from the Institutional Ethics Committee of Charité–Universitätsmedizin Berlin, was present (EA2/064/09), and the study was conducted according to the Declaration of Helsinki.

### Statistical Analysis

Metric variables are presented as medians (range) and categorical variables as frequencies. Group comparisons of continuous variables were performed using either the Mann–Whitney U test or the Kruskal–Wallis test, while the chi-squared test was used to compare categorical variables. Overall survival (OS), progression-free survival (PFS) and disease-free survival (DFS) were calculated with the Kaplan–Meier method. Log-rank tests were used to compare survival rates. OS was defined as the time from surgery to death, PFS as the time from surgery to first postoperative disease progression and DFS as the time from curative surgery to first postoperative disease recurrence. Patients who did not reach the respective endpoints or were lost to follow-up were censored at the last follow-up visit. Univariate and multivariate Cox’s regression analyses were performed to identify potential predictive factors for PFS and DFS. Results are given as a hazard ratio (HR) with a 95% confidence interval (95%-CI). *p* values less than 0.05 were considered statistically significant. The SPSS Statistics software, version 27 (IBM, Armonk, NY, USA), was used for statistical analyses.

## 3. Results

### 3.1. Patient Characteristics

A total of 161 patients with siNET were included, and the patients’ characteristics are displayed in Table 1. Curative resection was achieved in 53 (32.9%) cases. In the remaining 108 patients (67.1%), the reasons for incomplete resection of all tumor mass were the extent of distant metastases (*n* = 96, 88.9%), localization of central lymph node metastases (*n* = 11, 10.2%) and localization of locoregional lymph node metastases (*n* = 1, 0.9%). Lymphatic invasion was demonstrated in *n* = 105 (67.8%) patients, vascular invasion in *n* = 66 (43.1%) vascular and perineural invasion in *n* = 72 (65.5%).

The differences between siNET patients with and without L, V and Pn invasion are shown in Table 2 below. SiNET patients with L1, V1 and Pn1 status were more likely to have distant metastases at diagnosis (*p* = 0.005; *p* < 0.001; *p* < 0.001) and stage IV disease (*p* = 0.005; *p* < 0.001; *p* < 0.001), while they were less likely to undergo curative surgery (*p* < 0.001; *p* < 0.001; *p* < 0.001, respectively). Furthermore, patients with L, V and Pn invasion had larger tumors (T3 and T4) (*p* < 0.001; *p* < 0.001; *p* < 0.001), and they more often had lymph node metastases (*p* < 0.001, *p* = 0.037; *p* < 0.001) and positive resection margins (R1 and R2) (*p* = 0.004; *p* = 0.012; *p* < 0.001). Moreover, the amount of either L, V and Pn invasion was higher in patients with L1, V1 or Pn1, respectively. In addition, these patients had higher rates of postoperative progression (42.0% vs. 60.0%, *p* = 0.014, 41.4% vs. 66.7%, *p* = 0.003, 28.9% vs. 65.3%, *p* < 0.001). No differences were observed with respect to gender, age, grading, Ki67 and the occurrence of NET-related death.

### 3.2. Survival Analysis

As graphed in Figure 1a,c,e, patients with lymphatic, vascular or perineural invasion showed shorter progression-free survival (PFS) compared to those without (5-year PFS: L0: 64% vs. L1 40%; *p* = 0.005; V0 62% vs. V1 36%, *p* = 0.002 and Pn0: 71% vs. Pn1: 34%, *p* < 0.001). However, no differences in overall survival (OS) could be detected (s. Figure 1b,d,f).

### 3.3. Curative Surgery

When only focusing on the patients that received curative surgery (*n* = 53), 43.1% (*n* = 22) showed lymphatic, 20.8% (*n* = 11) showed microvascular and 29.7% (*n* = 11) showed perineural invasion. The respective characteristics of these patients are shown in Table 3 below.

The differences between siNET patients who underwent curative surgery with and without L, V and Pn invasion are shown in Table 4 below. Only the siNET patients with microvascular invasion were more likely to have distant metastases at diagnosis (2.4% vs. 45.5%, *p* < 0.001) and stage IV disease (2.4% vs. 45.5%, *p* < 0.001). Lymphatic invasion was associated with larger tumors (T3 and T4) (34.5% vs. 63.6%, *p* = 0.039), lymph node metastasis (44.8% vs. 90.09%, *p* < 0.001) and perineural invasion (10.3% vs. 31.8%, *p* < 0.001). Pn1 patients had higher rates of lymph node metastasis (46.2% vs. 90.9%, *p* = 0.011) and lymphatic (11.5% vs. 63.6%, *p* < 0.001) and microvascular invasion (3.8% vs. 54.5%, *p* < 0.001). Interestingly, patients with all L, V and Pn invasion had higher rates of disease recurrence (3.4% vs. 31.8%, *p* = 0.015, 7.1% vs. 45.5%, *p* = 0.006 and 0.0% vs. 36.4%, *p* = 0.005, respectively). Again, no differences were observed with respect to gender, age, grading, Ki67 and the occurrence of NET-related death.

### 3.4. Survival Analysis of siNET Patients Undergoing Curative Surgery

As graphed in Figure 2a,c,e, patients with lymphatic, vascular or perineural invasion who underwent curative surgery showed shorter disease-free survival (DFS) compared to those without (5-year DFS: L0: 96% vs. L1: 64%; *p* = 0.003; V0: 94% vs. V1: 38%, *p* < 0.001 and Pn0: 100% vs. Pn1: 59%, *p* < 0.001). However, no differences in overall survival (OS) could be detected between patients with or without L, V or Pn invasion (s. Figure 2b,d,f). When only focusing on stage I–III patients (*n* = 48), a shorter DFS was also confirmed in patients with LI (5-year DFS L0: 96% vs. L1: 64%, *p* = 0.004), VI (5-year DFS V0: 94% vs. V1: 29%, *p* < 0.001) and PnI (5-year DFS Pn0: 100% vs. Pn1: 58%, *p* = 0.001).

### 3.5. Cox’s Regression Model for Progression-Free and Disease-Free Survival

Univariate and multivariate Cox’s regression analyses were performed to further investigate the prognostic role of L, V and Pn invasion regarding progression- and disease-free survival. In the overall cohort, the univariate regression analysis, including the T, N and M (distant metastasis) stages as well as L, V, and PnI, identified all variables to be significantly associated with a higher risk of shorter PFS with hazard ratios above 1 (see Table 5). However, the multivariate analysis only revealed the T3 and 4 stages as well as distant metastasis as independent risk factors for PFS with hazard ratios of 2.584 (1.095–6.101), *p* = 0.030, and 3.022 (1.576–5.793), *p* < 0.001, respectively. Focusing only on the curative surgery subgroup, the univariate analysis identified the T3 and 4 stages and LI and VI as negative predictive factors for DFS (see Table 6). Therefore, these variables were included in the multivariate analysis, which revealed only vascular invasion as an independent negative risk factor for DFS.

## 4. Discussion

In the current study, we were able to show that siNET patients with tumors showing lymphatic, microvascular or perineural invasion had fewer curative surgeries, more distant metastases and higher rates of postoperative disease progression. Progression-free survival was significantly impaired in these patients, while overall survival was not. Focusing on siNET patients who underwent curative surgery, LI, VI and PnI were also associated with shorter disease-free survival but not with reduced OS. Moreover, VI was identified as an independent negative predictive factor for DFS.

In line with the existing literature, patients with siNET in this cohort faced a relatively good prognosis, although only about one-third of the patients (*n* = 53, 33.0%) were eligible for curative surgery. Several factors have been identified to influence disease recurrence, tumor progression and overall survival in siNET. For example, tumor stage, tumor grading, carcinoid disease and carcinoid heart disease are well-established risk factors for reduced overall survival [15,16,17,18]. Focusing on tumor recurrence and disease progression, multiple tumor manifestations, lymph node and distant metastases, tumor size and grading are confirmed negative predictors [19,20,21]. Consistent with previous findings [15,22], the risk for disease recurrence or progression in our cohort was lower when curative surgery could be achieved: while 54.0% of the whole cohort experienced postoperative disease progression, disease recurrence was observed in only 15.1% of the curatively resected patients. The identification and definition of risk factors for recurrence are of utmost priority for adequate patient-specific risk stratification as well as consequent treatment and surveillance regimens. In this context, we were able to show that, not only in the whole patient cohort but also in the curatively resected cohort, disease progression or recurrence occurred more often in the presence of lymphatic, microvascular or perineural invasion. LI, VI and PnI were also associated with impaired PFS and DFS. In addition, all three factors were determined to increase the risk for poor PFS in univariate regression analysis along with the T3 and 4 stages, N1 status and distant metastasis. On the other hand, in the multivariate analysis, none of LI, VI or PnI were identified as independent risk factors for poor PFS, while the T3 and 4 stages and distant metastasis were confirmed as such. The identification of tumor size and metastatic disease as negative predictive factors in siNET is in line with the existing literature [20,23]. L, V and Pn invasion were all associated with larger tumors and metastatic disease in the current study cohort. One could therefore carefully hypothesize that LI, VI and PnI were tumor characteristics with the potential for faster tumor proliferation as well as the development of (micro-)metastatic disease. However, exact interactions or causal links cannot be drawn from this retrospective observational study, and further research is needed to identify the underlying mechanisms.

Lymphatic, microvascular and perineural invasion are established risk factors for poor oncologic outcome in many solid tumors, including pancreatic, gastric and colorectal cancers [6,7,8,9]. In many studies, lymphatic and microvascular invasion are summarized as lymphovascular invasion (LVI) and are considered positive if malignant cells are detected within either the venous or the lymphatic spaces [24]. However, each type of invasion should be identified independently, as they may represent different ways of tumor spreading, i.e., hematogenous and lymphatic. In this context, in a recent study, Kiritani et al. found that patients with pancreatic neuroendocrine neoplasms who showed both lymphatic and vascular invasion had worse postoperative outcomes and higher rates of lymph node and liver metastases than patients with either LI or VI and patients without signs of LI or VI [24].

The identification of LI and VI via routine hematoxylin and eosin (H&E) staining can be challenging, and it therefore carries a risk of underestimation [11,25]. Hence, additional immunohistochemical staining methods have been introduced to increase detection rates, including, e.g., D2-40 and Elastica van Gieson staining [26,27,28]. However, the widespread use in routine histopathologic examination of NET specimens must be questioned, and inconsistencies between histopathologic examinations in the current literature must be kept in mind.

Evidence regarding the association between the histopathologic features of lymphatic, microvascular and perineural invasion and oncologic outcome in siNET is still limited. However, Kohno et al. investigated risk factors for the development of metastases in a small cohort of gastroenteropancreatic (GEP) NETs [29]. They found that microvascular invasion—above lymphatic invasion—was associated with both lymph node and distant metastases. Khetan et al. focused, in their study, on well-differentiated but advanced siNETs and demonstrated, similar to our findings, that perineural invasion was a risk factor for tumor progression in advanced siNET patients [30]. In another article, Manguso et al. analyzed a cohort limited to siNET patients who underwent surgery with curative intent [31]. Compared to the 15.1% recurrence rate in our curative surgery subgroup, they observed disease recurrence in 32%. They further found slightly higher rates of LVI (50.3%) and PnI (34%) than we found in our cohort, with LI in 43.1%, VI in 20.8% and PnI in 29.7%. Consistent with the presented findings, they also demonstrated that patients with recurrence were more likely to have LVI and PnI. While our data only identified microvascular invasion as an independent risk factor for disease recurrence, they additionally identified LVI and PnI along with mesenteric invasion by the primary tumor [31]. However, the presented sub-group of patients undergoing curative surgery is limited, with only 53 patients. Possible effects or associations of LI and PnI with DFS might be more evident in larger cohorts. Therefore, it remains likely that VI, and possibly also LI and PnI, might be potential indicators for micrometastatic disease already present at the time of surgery, which correlate with a higher risk of early disease progression and recurrence. Thus, we encourage a thorough and accurate histopathologic evaluation of siNET specimens so that the assessment of LI, VI and PnI becomes a routine part of the diagnostic workup. Furthermore, consideration of these tumor characteristics should be incorporated into interdisciplinary decision-making for patient-specific therapy and follow-up strategies.

Interestingly, there was no association between L, V and Pn invasion and the well-established predictive factor of tumor grading, suggesting an independent effect on disease recurrence and progression. In addition, we did not find any association between LI, VI and PnI and overall survival, which may not be totally surprising considering the overall good prognosis of siNET patients even in advanced and metastatic tumor stages.

Our study has some limitations. As per its retrospective nature, a selection bias cannot be excluded completely. Secondly, the limited size and the partly missing histopathologic data have to be mentioned and kept in mind when interpreting the results, especially with regards to perineural invasion. However, considering the rarity of the disease, this cohort with more than 160 included patients still represents a rather large study population. Due to the inclusion of patients from a relatively long study period, differences in the routine histopathologic assessment occurred, resulting in missing data when histologic parameters could not be retrieved from the medical records and the tumor specimens could not be re-evaluated anymore.

## 5. Conclusions

Lymphatic, microvascular and perineural invasion were associated with worse recurrence-free and disease-free survival in patients with siNET. Therefore, these features should be considered when making decisions about adjuvant therapy and follow-up regimens.

## Figures and Tables

**Figure 1 cancers-16-00305-f001:**
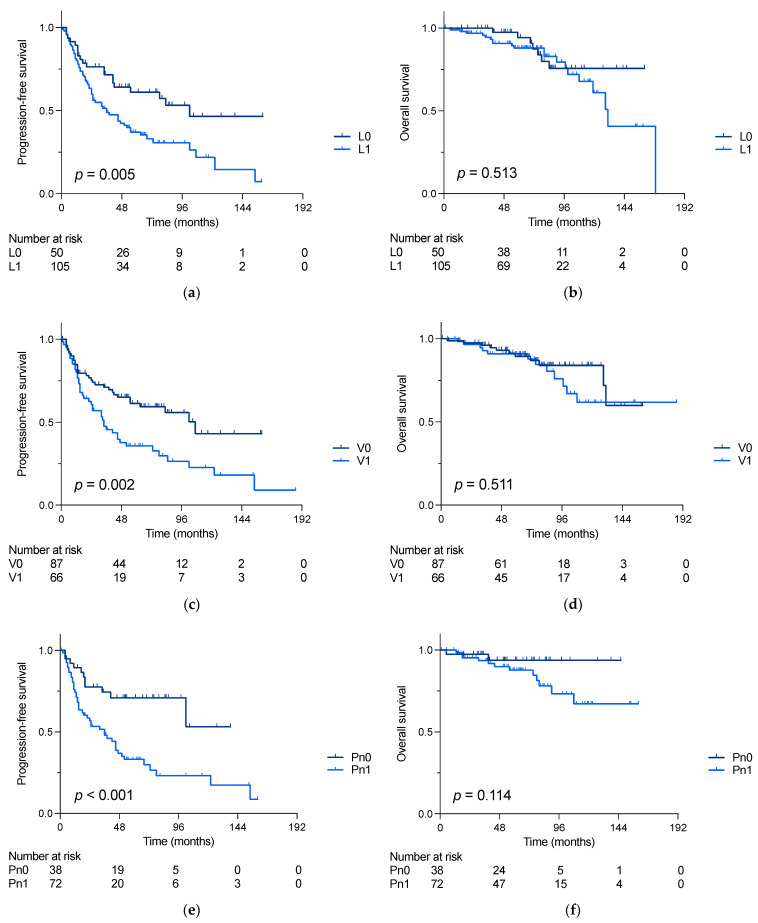
Progression-free (PFS) and overall survival (OS) of siNET patients undergoing small bowel resection. (**a**) PFS (5-year PFS: L0: 64% vs. L1 40%; *p* = 0.005) and (**b**) OS in patients comparing with and without lymphatic invasion (5-year OS L0: 92% vs. L1: 86%, *p* = 0.513); (**c**) PFS (5-year PFS V0: 61% vs. V1: 35%, *p* = 0.002) and (**d**) OS in patients comparing with and without vascular invasion (5-year OS V0: 90% vs. V1: 90%, *p* = 0.511); (**e**) PFS (5-year PFS Pn0: 75% vs. 30%, *p* < 0.001) and (**f**) OS in patients comparing with and without perineural invasion (5-year OS: Pn0 94% vs. Pn1: 87%, *p* = 0.114). Survival rates were compared using log-rank test.

**Figure 2 cancers-16-00305-f002:**
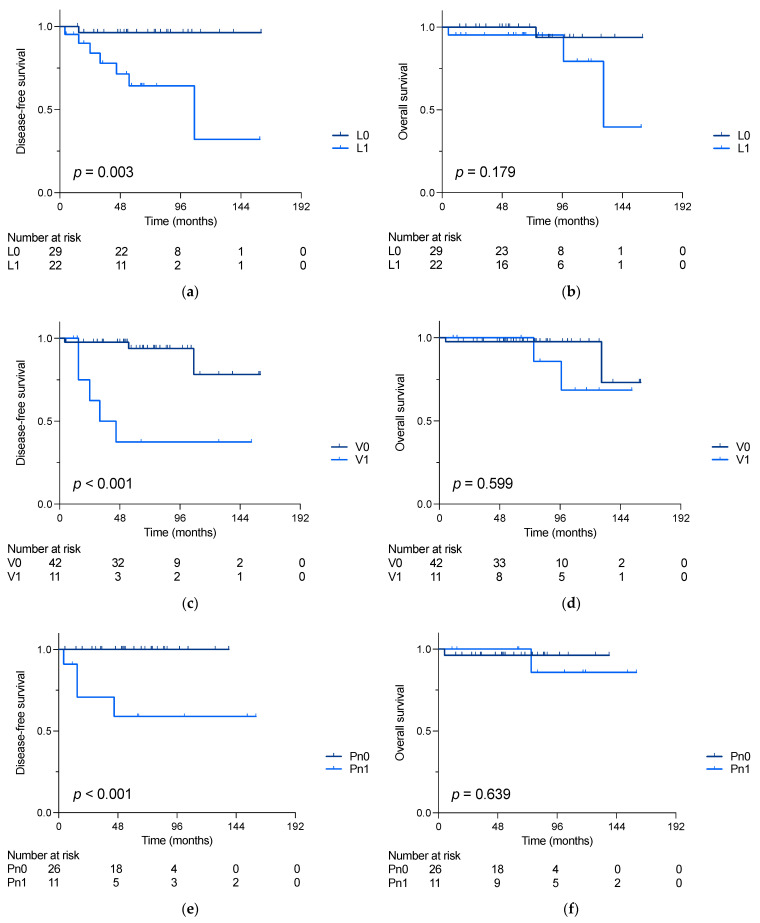
Disease-free (DFS) and overall survival (OS) of siNET patients undergoing small bowel resection with achieved curative resection. (**a**) DFS (5-year DFS: L0: 96% vs. L1: 64%, *p* = 0.003) and (**b**) OS in patients comparing with and without lymphatic invasion (5-year OS: L0: 100% vs. L1: 95%, *p* = 0.179); (**c**) DFS (5-year DFS: V0: 94% vs. V1: 38%, *p* < 0.001) and (**d**) OS in patients comparing with and without vascular invasion (5-year OS: V0: 97% vs. V1: 83%, *p* = 0.599); (**e**) DFS (5-year DFS: Pn0: 100% vs. Pn1: 59%, *p* < 0.001) and (**f**) OS in patients comparing with and without perineural invasion (5-year OS: Pn0: 96% vs. 86%, *p* = 0.639). Survival rates were compared using log-rank test.

**Table 1 cancers-16-00305-t001:** Clinicopathological characteristics of patients undergoing small bowel resection for small intestinal neuroendocrine tumors.

		*n* = 161
Sex ratio	Female:Male	94:67
Age (years) ^1^		62 (27–84)
Distant metastasis		102 (63.4%)
UICC stage	I	8 (5.0%)
	II	9 (5.6%)
	III	42 (26.1%)
	IV	102 (63.4%)
Curative surgery		53 (32.9%)
Histopathological Parameters		
T stage	T1	12 (7.5%)
	T2	26 (16.1%)
	T3	77 (47.8%)
	T4	46 (28.8%)
N stage	N0	22 (13.7%)
	N1	139 (86.3%)
Resection margins	R0	113 (70.2%)
	R1	37 (23.0%)
	R2	6 (3.7%)
	Rx	5 (3.1%)
Grading	G1	92 (57.1%)
	G2	69 (42.9%)
Lymphatic invasion ^2^	L0	50 (32.2%)
	L1	105 (67.8%)
Microvascular invasion ^3^	V0	87 (56.9%)
	V1	66 (43.1%)
Perineural invasion ^4^	Pn0	38 (34.5%)
	Pn1	72 (65.5%)
Follow-up		
Disease progression		87 (54.0%)
NET-related death		31 (19.3%)

Values in parentheses are percentages unless indicated otherwise; ^1^ median (range); ^2^ missing values: *n* = 6; ^3^ missing values: *n* = 8; ^4^ missing values *n* = 51.

**Table 2 cancers-16-00305-t002:** Comparison of siNET patients with and without lymphatic, vascular und perineural invasion who underwent bowel resection.

			L0 *n* = 50	L1 *n* = 105	*p*-Value	V0 *n* = 87	V1 *n* = 66	*p*-Value	Pn0 *n* = 38	Pn1 *n* = 72	*p*-Value
Sex ratio F:M			25:25	66:39	0.129	46:41	42:24	0.182	19:19	45:27	0.206
Age (years) ^1^			60 (27–78)	63 (29–84)	0.835	63 (27–80)	60 (32–84)	0.292	59 (27–78)	62 (30–84)	1.000
Distant metastases			24 (48.0%)	75 (71.4%)	0.005	41 (47.1%)	56 (84.8%)	<0.001	13 (34.2%)	61 (84.7%)	<0.001
UICC stage		I–III	26 (52.0%)	30 (28.6%)	0.005	45 (52.9%)	10 (15.2%)	<0.001	25 (65.8%)	11 (15.3%)	<0.001
		IV	24 (48.0%)	75 (71.4%)		42 (47.1%)	56 (84.8%)		13 (34.2%)	61 (84.7%)	
Curative surgery			29 (58.0%)	22 (21.0%)	<0.001	42 (48.3%)	11 (16.7%)	<0.001	26 (68.4%)	11 (15.3%)	<0.001
Histopathological Parameters											
T3/4			29 (58.0%)	90 (85.7%)	<0.001	57 (65.5%)	59 (89.4%)	<0.001	19 (50.0%)	67 (93.1%)	<0.001
N1			40 (60.0%)	103 (98.1%)	<0.001	70 (80.5%)	61 (92.4%)	0.037	23 (60.5%)	69 (95.8%)	<0.001
R1/2			6 (12.0%)	36 (34.3%)	0.004	16 (18.4%)	24 (36.4%)	0.021	3 (7.9%)	29 (40.3%)	<0.001
Grading	G1		29 (58.0%)	60 (57.1%)	0.920	48 (55.2%)	39 (59.1%)	0.628	24 (63.2%)	38 (52.8%)	0.297
	G2		21 (42.0%	45 (42.9%)		39 (44.8%)	27 (40.9%)		14 (36.8%)	34 (47.2%)	
Ki67 (%) ^1^			2 (1–17)	2 (1–20)	0.380	2 (1–18)	2 (1–20)	0.481	2 (1–17)	2 (1–20)	0.297
L1 ^2^			-	-	-	44 (50.6%)	54 (81.8%)	<0.001	8 (21.1%)	62 (86.1%)	<0.001
V1 ^3^			8 (16.0%)	54 (51.4%)	<0.001	-	-	-	2 (5.3%)	49 (68.1%)	<0.001
Pn1 ^4^			9 (18.0%)	62 (59.0%)	<0.001	22 (25.3%)	49 (74.2%)	<0.001	-	-	-
Follow-up											
Disease progression			21 (42.0%)	63 (60.0%)	0.014	36 (41.4%)	44 (66.7%)	0.003	11 (28.9%)	47 (65.3%)	<0.001
NET-related death			8 (16.0%)	21 (20.0%)	0.425	13 (14.9%)	14 (21.2%)	0.116	2 (5.3%)	14 (19.4%)	0.062

Values in parentheses are percentages unless indicated otherwise; ^1^ median (range); ^2^ missing values: *n* = 6; ^3^ missing values: *n* = 8; ^4^ missing values *n* = 51; F, female; M, male.

**Table 3 cancers-16-00305-t003:** Clinicopathological characteristics of patients undergoing curative surgery for small-intestinal neuroendocrine tumors.

		*n* = 53
Sex ratio	Female:Male	30:23
Age (years) ^1^		61 (27–80)
UICC	I	8 (15.1%)
	II	8 (15.1%)
	III	31 (58.5%)
	IV	5 (11.3%)
Distant metastases		6 (11.3%)
Histopathological Parameters		
T stage	T1	11 (20.8%)
	T2	18 (34.0%)
	T3	23 (43.4%)
	T4	1 (1.9%)
N stage	N0	18 (34%)
	N1	35 (66.0%)
Resection margins	R0	50 (94.3%)
	R1	2 (3.8%)
	R2	0
	Rx	1 (1.9%)
Grading	G1	33 (62.3%)
	G2	20 (37.7%)
Lymphatic invasion ^2^	L0	29 (56.9%)
	L1	22 (43.1%)
Microvascular invasion	V0	42 (79.2%)
	V1	11 (20.8%)
Perineural invasion ^3^	Pn0	26 (70.3%)
	Pn1	11 (29.7%)
Follow-up		
Disease recurrence		8 (15.1%)
NET-related death		4 (7.5%)

Values in parentheses are percentages unless indicated otherwise; ^1^ median (range); ^2^ missing values *n* = 2; ^3^ missing values *n* = 16.

**Table 4 cancers-16-00305-t004:** Comparison of patients undergoing curative surgery for siNET with and without lymphatic, vascular und perineural invasion.

		L0 *n* = 29	L1 *n* = 22	*p*-Value	V0 *n* = 42	V1 *n* = 11	*p*-Value	Pn0 *n* = 26	Pn1 *n* = 11	*p*-Value
Sex ratio		14:15	15:7	0.155	23:19	7:4	0.597	14:12	6:5	0.969
Age (years) ^1^		59 (27–78)	63 (41–80)	0.614	60 (27–78)	63 (49–80)	0.345	59 (27–78)	63 (30–80)	0.566
Distant metastases		4 (13.8%)	2 (9.1%)	0.688	1 (2.4%)	5 (45.5%)	<0.001	2 (7.7%)	3 (27.3%)	0.144
UICC stage	I–III	25 (86.2%)	20 (90.9%)	0.606	41 (97.6%)	6 (54.5%)	<0.001	24 (92.3%)	8 (72.7%)	0.144
	IV	4 (13.8%)	2 (9.1%)		1 (2.4%)	5 (45.5%)		2 (7.7%)	3 (27.3%)	
Histopathological Parameters										
T3/4		10 (34.5%)	14 (63.6%)	0.039	17 (40.5%)	7 (63.6%)	0.170	9 (34.6%)	7 (63.6%)	0.103
N1		13 (44.8%)	20 (90.9%)	<0.001	27 (64.3%)	8 (72.7%)	0.599	12 (46.2%)	10 (90.9%)	0.011
R1/2		1 (3.4%)	1 (4.5%)	0.842	1 (2.4%)	1 (9.1%)	0.299	0 (0%)	1 (9.1%)	0.119
Grading	G1	18 (62.1%)	14 (63.6%)	0.909	27 (64.3%)	6 (54.5%)	0.553	16 (61.5%)	8 (72.7%)	0.515
	G2	11 (37.9%)	8 (36.4%)		15 (35.7%)	5 (45.5%)		10 (38.5%)	3 (27.3%)	
Ki67 (%) ^1^		2 (1–10)	2 (1–12)	0.697	2 (1–12)	2 (1–6)	0.372	2 (1–10)	2 (1–12)	0.832
L1 ^2^		-	-	-	15 (35.7%)	7 (63.6%)	0.101	3 (11.5%)	7 (63.6%)	<0.001
V1		3 (10.3%)	7 (31.8%)	0.056	-	-	-	1 (3.8%)	6 (54.5%)	<0.001
Pn1 ^3^		3 (10.3%)	7 (31.8%)	<0.001	5 (11.9%)	6 (54.5%)	0.002	-	-	-
Follow-up										
Disease recurrence		1 (3.4%)	7 (31.8%)	0.015	3 (7.1%)	5 (45.5%)	0.006	0 (0.0%)	4 (36.4%)	0.005
NET-related death		1 (3.4%)	3 (13.6%)	0.118	2 (4.8%)	2 (18.2%)	0.253	1 (3.8%)	1 (9.1%)	0.761

Values in parentheses are percentages unless indicated otherwise; ^1^ median (range); ^2^ missing values *n* = 2; ^3^ missing values *n* = 16; F, female; M, male.

**Table 5 cancers-16-00305-t005:** Uni- and multivariate Cox’s regression analysis for variables affecting progression-free survival in siNET patients.

	Univariate		Multivariate	
	HR (95%-CI)	*p* Value	HR (95%-CI)	*p* Value
T3/4	5.591 (2.573–12.150)	<0.001	2.584 (1.095–6.101)	0.030
N1	3.404 (1.480–7.830)	0.004	1.598 (0.605–4.224)	0.344
L1	1.197 (1.196–3.236)	0.008	0.867 (0.440–1.707)	0.679
V1	1.968 (1.265–3.063)	0.003	1.055 (0.596–1.867)	0.854
Pn1	3.106 (1.607–6.002)	<0.001	1.281 (0.519–3.163)	0.591
Distant metastasis (M1)	4.847 (2.728–8.611)	<0.001	3.022 (1.576–5.793)	<0.001

HR, hazard ratio; 95%-CI, 95% confidence interval.

**Table 6 cancers-16-00305-t006:** Uni- and multivariate Cox’s regression analysis for variables affecting disease-free survival in patients undergoing curative surgery for siNET.

	Univariate		Multivariate	
	HR (95%-CI)	*p* Value	HR (95%-CI)	*p* Value
T3/4	10.762 (1.319–87.803)	0.027	4.129 (0.421–40.456)	0.223
N1	1.924 (0.377–9.818)	0.431		
L1	12.250 (1.502–99.924)	0.019	4.820 (0.473–49.112)	0.184
V1	0.479 (2.225–40.380)	0.002	7.716 (1.579–37.700	0.012
Pn1	350,371.244 (0.0–7.912 × 10^157^)	0.943		
Distant metastasis (M1)	1.489 (0.182–12.183)	0.711		

HR, hazard ratio; 95%-CI, 95% confidence interval.

## Data Availability

Data are contained within the article.

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
