# Peer review of "Influence of Lymphatic, Microvascular and Perineural Invasion on Oncological Outcome in Patients with Neuroendocrine Tumors of the Small Intestine"

_cancers, 2024, doi:10.3390/cancers16020305_

Round 1
Reviewer 1 Report
Comments and Suggestions for Authors
Thank you for an interesting and well-written manuscript. It has a lot of potential and deserves publication if my comments are answered.
1. Why did you include NET G1 and 2, nut exclude NET G3? Would it not have been more reasonable to include NET G3 and only exclude NECs?
2. How do we know that it is lymphovascular/perineural invasion which is the cause of the poor prognosis and not the presence of distant metastases? There is no Cox-regression? Please perform that with inclusion of TNM stage so the impact can be demonstrated further.
3. Could you sepculate on why only 1/3 of the patients are eligeble for surgery?
4. Pi is only evaluated in 110 of 161 patients. Is it possible to re-examine the rest of the tissue?
Reviewer 2 Report
Comments and Suggestions for Authors
This is a retrospective analysis of the impact of lymphatic, venous and perineurial invasion in small intestine NET. The subject is of clinical interest for sm NET and clinical research is limited. The study population is quite high and results and conclusions are presented reasonable.
However, there is a shortcoming in presenting the study population included in this analysis.
In concrete, there are 510 pts included in the CCC of the Charite, of whom 161 are included in this manuscript. The reason for exclusion should be given in detail and in numbers (for example xxx because of G3...)
While 53 pts were classified as curative surgery, additionally the reason for non curative surgery should be given for the 108 patients with non curative surgery
In Table 1, some numbers are unclear and do not sum up to 161 pts, for example UICC Stage, resection margin, T stage. If this data are missing, this should be noted
